# Pharmacokinetics, feasibility and safety of co-administering azithromycin, albendazole, and ivermectin during mass drug administration: A review

Scott McPherson[1,2], Anthony W. Solomon[1,3], Fikre Seife[4], Hiwot Solomon[4], Teshome Gebre[5], David C. W. Mabey[1], Michael Marks[1,6,7]*

1 Clinical Research Department, Faculty of Infectious and Tropical Diseases, London School of Hygiene & Tropical Medicine, London, United Kingdom, 2 RTI International, Research Triangle Park, North Carolina, United States of America, 3 Global Neglected Tropical Diseases Programme, World Health Organization, Geneva, Switzerland, 4 Federal Ministry of Health, Addis Ababa, Ethiopia, 5 International Trachoma Initiative, Task Force for Global Health, Addis Ababa, Ethiopia, 6 Hospital for Tropical Diseases, University College London Hospital, London, United Kingdom, 7 Division of Infection & Immunity, University College London, London, United Kingdom

* Michael.marks@lshtm.ac.uk

**Data Availability Statement:** All data are available within the manuscript.

## Abstract

### Introduction

Traditionally, health ministries implement mass drug administration programmes for each neglected tropical disease (NTD) as separate and distinct campaigns. Many NTDs have overlapping endemicity suggesting co-administration might improve programme reach and efficiency, helping accelerate progress towards 2030 targets. Safety data are required to support a recommendation to undertake co-administration.

### Methodology

We aimed to compile and summarize existing data on co-administration of ivermectin, albendazole and azithromycin, including both data on pharmacokinetic interactions and data from previous experimental and observational studies conducted in NTD-endemic populations. We searched PubMed, Google Scholar, research and conference abstracts, gray literature, and national policy documents. We limited the publication language to English and used a search period from January 1st, 1995 through October 1st, 2022. Search terms were: azithromycin and ivermectin and albendazole, mass drug administration co-administration trials, integrated mass drug administration, mass drug administration safety, pharmacokinetic dynamics, and azithromycin and ivermectin and albendazole. We excluded papers if they did not include data on co-administration of azithromycin and both albendazole and ivermectin, or azithromycin with either albendazole or ivermectin alone.

### Results

We identified a total of 58 potentially relevant studies. Of these we identified 7 studies relevant to the research question and which met our inclusion criteria. Three papers analyzed

**Funding:** The author(s) received no specific funding for this work.

**Competing interests:** I have read the journal's policy and the authors of this manuscript have the following competing interests: AWS is a staff member of the World Health Organization.

pharmacokinetic and pharmacodynamic interactions. No study found evidence of clinically significant drug-drug interactions likely to impact safety or efficacy. Two papers and a conference presentation reported data on the safety of combinations of at least two of the drugs. A field study in Mali suggested the rates of adverse events were similar with combined or separate administration, but was underpowered. A further field study in Papua New Guinea used all three drugs as part of a four-drug regimen also including diethylcarbamazine; in this setting, co-administration appeared safe but there were issues with the consistency in how adverse events were recorded.

## Conclusion

There are relatively limited data on the safety profile of co-administering ivermectin, albendazole and azithromycin as an integrated regimen for NTDs. Despite the limited amount of data, available evidence suggests that such a strategy is safe with an absence of clinically important drug-drug interactions, no serious adverse events reported and little evidence for an increase in mild adverse events. Integrated MDA may be a viable strategy for national NTD programmes.

## Author summary

Treatment of the whole community (mass drug administration, MDA) has been a major intervention strategy against many neglected tropical diseases (NTDs) over the last decade. Normally health ministries deliver individual MDA rounds targeting specific NTDs. This multiplies the training, transport and time burden for local health service personnel in districts in which several NTDs are present, imposing considerable financial and human resource costs to health ministries and their partners, and causing requiring repeated disruption to the daily life of communities receiving MDA. Delivering MDA for several NTDs at one time could improve the efficiency of NTD programmes. We reviewed existing data on the safety and feasibility of combining MDA of albendazole, ivermectin and azithromycin into a single co-delivered MDA. Several studies had evaluated if taking these drugs at the same time changed drug levels in recipients' blood; these studies concluded that there was not an important difference in blood drug levels comparing instances when the medicines were taken separately to instances when they were taken at the same time. Two non-randomised studies assessed side effects experienced by people taking combinations of the three drugs and suggested doing so was safe. One small study in Mali had assessed combining all three drugs and also suggested this was safe but was too small to give a definitive answer. Two studies in Papua New Guinea assessed all three drugs being taken together in combination with a fourth drug, diethylcarbamazine. These studies also suggest co-administration was safe overall. Most of the identified studies had some methodological shortcomings, such as small sample sizes or issues with the way adverse events were recorded. Overall, the data suggest co-administration of azithromycin, ivermectin and albendazole is viable, but larger safety studies are needed.

## Background

Neglected tropical diseases (NTDs) are a major cause of preventable illness and death in low- and middle- income countries [1]. Through a combination of improved access to water,

sanitation and hygiene; intensified disease management; mass drug administration (MDA); vector control; and veterinary public health, many countries have either eliminated or are on track to eliminate one or more NTDs. The recent road map document "Ending the neglect to attain the Sustainable Development Goals: A road map for neglected tropical diseases 2021–2030" sets out global targets and milestones to prevent, control, eliminate and eradicate 20 diseases and disease groups by 2030. The road map is built on three interlinked pillars of accelerating programmatic action, intensifying the use of cross-cutting approaches, and switching to operating models that facilitate country ownership [2].

MDA involves offering treatment to all members of a community or population within a defined area, without individual-level diagnostic testing and regardless of whether or not any specific individual has the targeted disease or infection. Three NTDs that are aiming to achieve global elimination targets using MDA as a critical part of the intervention strategy are lymphatic filariasis, onchocerciasis and trachoma.

Confidence in the safety of these drugs combined is bolstered by a long history of mass drug administration among endemic communities. Ivermectin has been consistently used at scale since the Mectizan Donation Program began donating Ivermectin in 1987 [3] administered alone or together with albendazole. Azithromycin is a commonly used antibiotic that has also been used in trachoma MDA campaigns throughout the world. Collectively many hundred million doses of each agent have been delivered as part of NTD programmes.

MDA of ivermectin and albendazole is also effective against other NTDs, including soil-transmitted helminthiases, scabies and strongyloidiasis. MDA of azithromycin is also effective against yaws [4–7]. Subsets of these diseases can therefore be managed with a different medicine or combination of medicines in annual or bi-annual MDA campaigns; the main drivers of community-based MDA programmes that employ albendazole, azithromycin and ivermectin are currently lymphatic filariasis, onchocerciasis and trachoma. For trachoma, a number of rounds of antibiotic MDA are recommended, with the antibiotic being oral azithromycin for most individuals and 1% tetracycline eye ointment for those unable or unwilling to take azithromycin; the number of rounds indicated is dependent on the prevalence of trachomatous inflammation—follicular (TF) in children aged 1–9 years [8]. For onchocerciasis, the second leading infectious cause of blindness behind trachoma, WHO recommends annual or biannual treatment with ivermectin for at least 14 rounds [9]. Lymphatic filariasis is treated with a variety of MDA combinations, including ivermectin and albendazole, or diethylcarbamazine (DEC) and albendazole, for at least five years [10]. In some specific situations, a three-drug combination of albendazole, DEC and ivermectin is recommended but not in settings in which onchocerciasis is co-endemic [11].

Following the NTD road map's first pillar of 'accelerating progress', national NTD programmes are exploring ways to improve upon standard disease-specific MDA strategies. MDA requires an enormous effort from every tier of a national health system to train the distribution workforce, sensitize targeted communities, deliver medicines through the national supply chain, administer medication, and collate treatment reports. In order to achieve elimination targets by 2030, ease strain on country health systems and adapt to shifting availability of resources, there is interest in combining individual drug regimens into co-administered MDA packages. Co-administered MDA could reduce the financial cost and time required on the part of health workers, as well as reduce the accompanying loss of their attention to other health initiatives. Communities could also benefit by reduced disruption to their daily activities if interventions are delivered simultaneously. Both communities and health systems may benefit if this translates into increased coverage of interventions. As lymphatic filariasis is treated through much of its endemic area with the combined regimen of ivermectin and albendzole, onchocerciasis programmes can already be integrated. However, adding in

azithromycin MDA (for trachoma or yaws) could further benefit NTD programmes in many countries.

As part of a larger project on integrated MDA in Ethiopia [12] we conducted a review of existing data on the safety and feasibility of co-administration of ivermectin, albendazole and azithromycin, to evaluate the potential role of combination MDA of these three medicines in accelerating progress towards 2030 targets.

## Methodology

We aimed to compile and summarize existing data on co-administration of ivermectin, albendazole and azithromycin, including both data on pharmacokinetic interactions and data from previous experimental and observational studies conducted in NTD-endemic populations. We used the 'PICOT' method (BOX 1) for formally framing the question that we intended the review to address. The final question derived was "When azithromycin, albendazole and ivermectin are combined during mass drug administration, is there a documented significant increase in adverse events as compared to giving azithromycin separately?"

We searched PubMed, Google Scholar, research and conference abstracts, gray literature, and national policy documents. We limited the publication language to English and used a search period from January 1st, 1995 through October 1st, 2022. Search terms were: azithromycin and ivermectin and albendazole, mass drug administration co-administration trials, integrated mass drug administration, mass drug administration safety, pharmacokinetic and azithromycin and ivermectin and albendazole. We excluded papers if they did not include azithromycin and both albendazole and ivermectin, or azithromycin with either albendazole or ivermectin alone.

We identified relevant papers by screening the abstracts. For relevant papers, a standardized data extraction form was used to record data on eligibility, methods, participants, intervention groups, outcome measures and results. For the purpose of analysis, studies were grouped into pharmacokinetic studies and field evaluations and by the specific drug combinations evaluated.

## Results

We identified a total of 66 potentially relevant papers (Fig 1). Of these, we identified 7 studies that were relevant to the research question and met our inclusion criteria. Three papers

---

### Box 1: PICOT approach

- **P**opulation: patients living in endemic districts receiving MDA

- **I**ntervention: co-administered albendazole, azithromycin, and ivermectin

- **C**omparison: these drugs administered together or separately

- **O**utcomes: the incidence of serious adverse events and all adverse events

- **T**rial: experimental or observational studies

- The **T**rial is a Randomized controlled trial

---

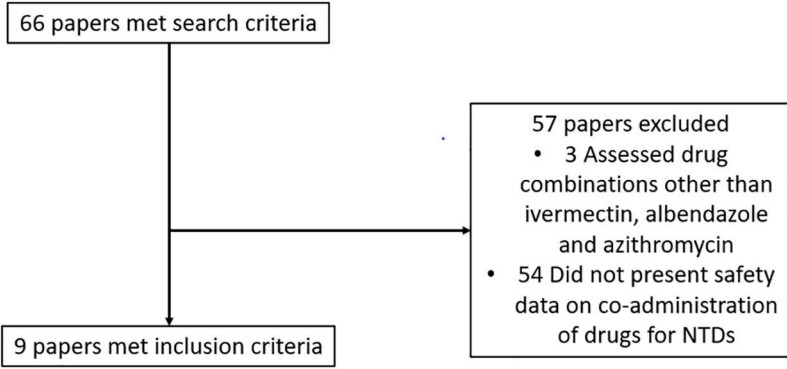

**Fig 1. Search results for papers included in the review.**

examined potential pharmacokinetic and pharmacodynamic interactions of azithromycin, ivermectin and albendazole. Two papers and a conference presentation reported data on co-administration of at least two of the three drugs. One paper was a field study that involved the co-administration of all three drugs. A further field study reported on the co-administration of albendazole, azithromycin, and ivermectin as part of a four-drug regimen that also included diethylcarbamazine.

## Pharmacokinetic studies

Previous pharmacokinetic studies demonstrated that there is little to no relevant drug-drug interaction between ivermectin and albendazole and these data have underpinned the co-administration of these drugs for lymphatic filariasis [13,14,15]. Mass co-administration of both drugs to treat lymphatic filariasis has occurred with no reported serious adverse events related to drug interactions among large populations for many years [16]. We identified three additional pharmacokinetic studies related to the interaction of the three-drug combination including azithromycin.

The first study was a randomized, three-way crossover pharmacokinetic study of potential interactions between azithromycin, ivermectin and albendazole [13]. In this study, 18 volunteers were administered a fixed dose of 500 mg of azithromycin alone, a fixed dose of 400 mg of albendazole alone, and a dose proportional to body weight (200 ug/kg rounded to the nearest 3 mg) of ivermectin alone. All three drugs were administered together. A total of 19 blood samples were collected from each subject before the administration took place and then at intervals over a 168-hour period. The study measured total exposure to each drug over time ($AUC_{0-t}$) and peak concentrations of the drug after dosing ($C_{max}$). When all three drugs were combined, the azithromycin $AUC_{0-t}$ and $C_{max}$ were increased approximately 13% and 20%, the albendazole $AUC_{0-t}$ was decreased approximately 3%, the albendazole $C_{max}$ was increased approximately 3%, and the ivermectin $AUC_{0-t}$ and $C_{max}$ were increased approximately 31% and 27%, respectively [13]. Albendazole sulfoxide, the active metabolite after absorption, decreased when both drugs were administered together compared to when albendazole was administered alone. These changes in drug levels were felt to be small and unlikely to be of clinical importance. The elevation in ivermectin levels was within established safety ranges for the drug [14].

The second study was a pharmacokinetic modeling exercise to explore the mechanisms of interaction when azithromycin is combined with ivermectin and albendazole, with a particular focus on the AUC for ivermectin [15]. The authors constructed two simulated

pharmacokinetic models to evaluate the impact of co-administration and other variables on ivermectin levels. One thousand interaction studies were simulated to explore extreme high ivermectin values that might occur. Using both models, the predicted highest ivermectin concentrations were 115–201 ng/mL: well inside the established safety range.

A third pharmacokinetic study was conducted in Papua New Guinea. This study was a three-arm pharmacokinetic interaction study in which 36 volunteers were recruited and randomized to either a) ivermectin (150 μg/kg), diethylcarbamazine (6mg/kg) and albendazole (IDA) (400mg); b) IDA combined with azithromycin (30mg/kg to a maximum dose of 2gm); or c) azithromycin alone. Drugs were administered to participants after a night of fasting. Drug levels were measured via repeated sampling between baseline and 72 hours. Drug levels were evaluated using mass spectrometry and reported as Cmax and AUC [17]. Total drug exposure was unaffected by co-administration when compared to the stand-alone AUC of each drug. The AUC during co-administration for ivermectin was reduced to 87.9%, for DEC to 92.9%, while albendazole's AUC was unaffected. In this small study no difference was seen in the number of adverse events between study arms.

## Observational studies

One observational study was identified examining the combination of albendazole and azithromycin and one observational study examining the combination of azithromycin and ivermectin. In Colombia, the Ministry of Health adopted a policy of co-administering albendazole (400mg) for the treatment of soil-transmitted helminths alongside azithromycin (20mg/kg to a maximum of 1gm) given for the elimination of trachoma. These activities were carried out programmatically with safety monitoring done in a sub-group of recipients. More than 300,000 people received albendazole and azithromycin, taken together, over three separate annual cycles of MDA from 2012–14. Adverse events were ascertained in a subgroup of 4,438 individuals. Adverse events were reported in 0.16%, of which the most common were headache, dizziness and diarrhoea. No serious adverse events were documented [18].

In the Solomon Islands, a team conducted a prospective, single-arm, intervention study to assess the safety and efficacy of combined MDA of ivermectin (200ucg/kg) and azithromycin (20mg/kg to a maximum of 1gm). Azithromycin was used to treat trachoma while ivermectin was used to treat scabies. Overall, 21,181 individuals received the combination treatment. Safety data were collected via a questionnaire administered both at baseline and one week following MDA. In most villages the questionnaire was administered by the routine health service whilst in ten villages the questionnaire was administered by a dedicated research team. There were no reported serious adverse events. Overall, 2.6% of the entire study population reported an adverse event. In the ten villages visited by a research team 4.1% of participants reported an adverse event. All adverse events reported were mild and short-lived. Gastrointestinal symptoms, dizziness and itch were the most commonly reported adverse events, in keeping with the known safety profile of the drugs. Passive surveillance in the 12 months before and after MDA showed that the number of hospital admissions (1530 vs 1602) and deaths (73 vs 83) were similar before and after MDA [19,20,21].

## Randomised controlled trials

The AZIVAL study was conducted in Mali in four villages (total population 3, 011) endemic for trachoma and lymphatic filariasis. Two villages were randomized to co-administration of ivermectin, albendazole, and azithromycin and two villages received an initial round of MDA of ivermectin (150mcg/kg) and albendazole (400mg) followed one week later by a round of azithromycin MDA (20mg/kg to a maximum of 1gm). Safety data were collected by the study

team from all participants on days 1, 8 and 15 after treatment, via clinical examination and adverse event questionnaire. The overall reported rates of any adverse event were similar in the co-administration arm (281/1501, 18.7%) and the standard treatment arm (239/1510, 15.8%). No serious adverse events were reported. The most frequent adverse events were abdominal pain, headache and diarrhoea, in line with the drugs' established safety profiles [22].

The final identified study was an open-label, cluster-randomized trial conducted in two study sites, Namatanai and Lihir Island, Papua New Guinea. Clusters were randomised to receive either MDA of ivermectin (150mcg/kg), diethylcarbamazine (6mg/kg) and albendazole (400mg) (IDA) followed by a single dose of azithromycin (30mg/kg to a maximum of 2gm) administered one week later, or MDA of all four drugs together. Data on adverse events were collected in the 24–48 hours following MDA. Study teams initially used a general question concerning adverse events within both study arms in Namatanai. However, because the researchers noticed a low proportion of reported adverse events, on Lihir Island, a more in-depth questionnaire concerning specific adverse events was used. Overall, 7,281 people received the four-drug regimen and no serious adverse events occurred. In clusters that received separate MDA, the rate of adverse events was 6.3% following IDA and 9.9% following azithromycin. In clusters that received combined MDA, the rate of adverse events was 6.9%. The incidence of reported adverse events was higher when the more detailed questionnaire was used. The most commonly reported adverse events were fever, headache and abdominal pain [23].

## Discussion

Our review highlights the relative paucity of data on the safety profile of co-administering ivermectin, albendazole and azithromycin as an integrated regimen for NTDs. However, the available evidence suggests that co-administration is safe, with an absence of clinically important drug interactions, no serious adverse events reported to date amongst tens of thousands of recipients of combined treatment, and little evidence for an increase in more mild adverse events. Overall, our review suggests that integrated MDA is a viable strategy for national programmes.

Whilst the published pharmacokinetic studies are small, they have consistently demonstrated either an absence of any drug interaction, or that the alterations in drug levels when these agents are co-administered are unlikely to impact drug efficacy or to be of concern from a safety perspective. Multiple field studies of different drug combinations have been conducted in diverse geographic regions and in settings in which a range of NTDs are endemic. The most commonly reported adverse events are in keeping with the known safety profiles of the three drugs. The absence of serious adverse events and the absence of a significant increase in the frequency of mild adverse events are very encouraging from a safety perspective.

We identified a number of limitations in the existing literature. None of the relevant papers included a pediatric population. While the available research suggests extrapolation of results from adults to children may be possible, further studies specifically focused on children would be an advantage for large scale coadministration expansion. The AZIVAL study, conducted in Mali, had a suitable study design but too small a sample size to definitively answer the question on safety and justify a programmatic recommendation for co-administration [22]. The study conducted in the Solomon Islands, relied on before-and-after comparisons rather than being randomized [21]. It also was undertaken in a setting which was not endemic for onchocerciasis or lymphatic filariasis. As adverse events can be related to parasite burden, this may affect the generalisability of the findings. Although Colombia has conducted large scale integrated

MDA, systematic safety data are only available from a small proportion of recipients in those campaigns. The largest co-administration trial to date, in Papua New Guinea, ended with a smaller number of people treated than originally targeted and the investigators altered the data collection method for adverse events during the study [23]. Given the ultimate goal of trials involving the co-administration of these drugs, the studies found in the existing literature have not achieved the sample size that WHO has previously used for guideline adoption in similar co-administration trials of different drug combinations [24]. Collectively these data highlight the need for larger, more robust field studies in settings where onchocerciasis, lymphatic filariasis and trachoma are co-endemic, to provide greater certainty for national programmes interested in adopting integrated MDA.

Our review also has a number of limitations. We focused on published literature and may have excluded more real-world programmatic experience involving the co-administration of ivermectin, albendazole, and azithromycin. Alongside Columbia, at least anecdotally, other countries have conducted MDA using combinations of these drugs, but we were unable to identify any published safety data from those experiences. Our review focused on one particular three-drug combination, but alternative triple-drug approaches (such as the combination of ivermectin, albendazole and praziquantel) have been used in other settings [25]. Whilst such studies may not directly inform the safety profile of the combination under review here, they may provide important insights including how integrated MDA can result in higher coverage and cost savings. Given the relative a paucity of data on the co-administration of these three drugs, it may also be relevant to also investigate whether administration with or without food further impacts drug levels [26–28]. Finally, as noted above, there are limitations with the design and interpretation of studies included in our review.

We found multiple studies suggesting that co-administration may be a safe approach, conferring a risk of adverse events that is similar to MDA of the different drugs delivered separately. The data support the conduct of larger, well-powered trials using a standardised safety monitoring approach, to generate the evidence needed to support global adoption of co-administration.

## Acknowledgments

### Disclaimer

The authors alone are responsible for the views expressed in this article and they do not necessarily represent the views, decisions or policies of the institutions with which they are affiliated.

## Author Contributions

**Conceptualization:** Scott McPherson, Anthony W. Solomon, Fikre Seife, Hiwot Solomon, Teshome Gebre, David C. W. Mabey, Michael Marks.

**Data curation:** Scott McPherson.

**Formal analysis:** Scott McPherson.

**Supervision:** David C. W. Mabey, Michael Marks.

**Writing – original draft:** Scott McPherson.

**Writing – review & editing:** Anthony W. Solomon, Fikre Seife, Hiwot Solomon, Teshome Gebre, David C. W. Mabey, Michael Marks.

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
