## [Decision Letter · Decision Letter 0]

24 Apr 2023

Dear Dr. Marks,

Thank you very much for submitting your manuscript "Pharmacodynamics, feasibility and safety of co-administering azithromycin, albendazole, and ivermectin during mass drug administration: a review" for consideration at PLOS Neglected Tropical Diseases. As with all papers reviewed by the journal, your manuscript was reviewed by members of the editorial board and by several independent reviewers. The reviewers appreciated the attention to an important topic. Based on the reviews, we are likely to accept this manuscript for publication, providing that you modify the manuscript according to the review recommendations. 

Two highly qualified reviewers find merit in the manuscript but each identifies some areas in which improvement is warranted, as specified in detail in their comments. These are suggestions for minor improvement but are both relevant and justified. The authors need to address each in detail in a letter accompanying submission of a revised version of the manuscript. I believe that this represents a significant addition to the literature and will be of interest to the scientific community in this field.

Sincerely,

Timothy G. Geary, PhD

Guest Editor

Mathieu Picardeau

Section Editor

Two highly qualified reviewers find merit in the manuscript but each identifies some areas in which improvement is warranted, as specified in detail in their comments. These are suggestions for minor improvement but are both relevant and justified. The authors need to address each in detail in a letter accompanying submission of a revised version of the manuscript. I believe that this represents a significant addition to the literature and will be of interest to the scientific community in this field.

Reviewer's Responses to Questions

**Key Review Criteria Required for Acceptance?**

**Methods**

-Are the objectives of the study clearly articulated with a clear testable hypothesis stated?

-Is the study design appropriate to address the stated objectives?

-Is the population clearly described and appropriate for the hypothesis being tested?

-Is the sample size sufficient to ensure adequate power to address the hypothesis being tested?

-Were correct statistical analysis used to support conclusions?

-Are there concerns about ethical or regulatory requirements being met?

Reviewer #1: This manuscript represents a review of information available in the public domain with regard to the pharmacokinetics and safety of albendazole, ivermectin, and azithromycin. The article is not a research article per se; accordingly, there is not a testable hypothesis for this review. The authors correctly identify that there is a paucity of information available based on their search criteria but do not identify next steps for study(s) that could be done to impact MDA policy change.

Reviewer #2: see comments

**Results**

-Does the analysis presented match the analysis plan?

-Are the results clearly and completely presented?

-Are the figures (Tables, Images) of sufficient quality for clarity?

Reviewer #1: As noted in methods, there was no analysis plan in this review. No figures (tables/images) are presented.

Reviewer #2: yes, could include table on doses and side effects

**Conclusions**

-Are the conclusions supported by the data presented?

-Are the limitations of analysis clearly described?

-Do the authors discuss how these data can be helpful to advance our understanding of the topic under study?

-Is public health relevance addressed?

Reviewer #1: See summary and general comments below

Reviewer #2: (No Response)

**Editorial and Data Presentation Modifications?**

Reviewer #1: Minor comments:

Since there are no pharmacodynamic analyses reviewed or presented, the title should be revised accordingly.

Similarly, in the methods section of the abstract and body of the manuscript, the word “dynamics” should be deleted.

The pharmacokinetic studies cited were conducted in adults. The authors should indicate whether studies in the pediatric population would be necessary, or whether the results in adults could be extrapolated to the pediatric population.

Please double check reference 11. Chuck Knirsch has been inadvertently been left off the reference.

Reviewer #2: appropriate, minor revisioin

**Summary and General Comments**

Reviewer #1: This manuscript represents a review of information available in the public domain with regard to the pharmacokinetics and safety of albendazole, ivermectin, and azithromycin. In the process of reviewing the literature, the authors discovered that there is indeed a paucity of information available for these three drugs administered in combination yet concluded that integrated MDA with this combination may be a viable strategy for national NTD programmes.

Although the conclusion is based on triple-drug combination, the authors failed to capture a key component of confidence in safety; that is, the number of exposures over time for each individual component of the combination. Ivermectin has been used widely in MDA since 1989 alone or in combination with albendazole. Albendazole has been widely used since the 1980s in community-based treatment paradigms either alone or in combination with other drugs. Azithromycin is a widely used antibiotic for multiple indications that has also been used in trachoma MDA campaigns. Although there may be a paucity of information on the three drugs used in combination as single doses, our confidence in safety is large given the millions of doses administered worldwide over the last 30+ years. The same would not be true for new chemical entities used in combination. I would suggest that the authors revisit their discussion about confidence in safety along these lines.

With regard to pharmacokinetics, the authors cite the relevant studies available in the scientific literature. Having said that, there was mention of the analyte albendazole but not the active metabolite albendazole sulfoxide. Albendazole administered orally is poorly bioavailable, partly through pH-dependent solubility mitigated in part by the administration of food, but also because albendazole parent is rapidly metabolized (likely in both gut and liver by CYP3A4) to albendazole sulfoxide. In fact, following a single 400 mg dose of albendazole, it is difficult to measure albendazole parent and the bioactive metabolite is indeed the more important analyte. In one of the pharmacokinetic studies cited (Amsden et al), there was virtually no change in albendazole parent when administered with azithromycin, whereas there was a decrease in albendazole sulfoxide when administered in combination. Although the decrease is not likely clinically relevant, it illustrates the point that one should take into account the relevant analyte, in this case the active metabolite of this anthelminthic drug albendazole.

There is a known food effect for both albendazole and ivermectin exposures. The authors should take this into account when reviewing the pharmacokinetic and safety data and discuss whether food should be recommended when the three drugs are used in combination.

I would agree with the authors that there is unlikely to be a clinically-relevant drug-drug pharmacokinetic interaction with albendazole, ivermectin and azithromycin in combination. Based on the wide-use of these drugs individually, in two-drug combinations (ie ivermectin and albendazole), or in triple drug combination where yaws or trachoma are prevalent, the likelihood of a significant novel safety finding – especially with regard to serious adverse events – is unlikely to surface. Having said that, if the triple combination is being considered for WHO guidelines review, the standard practice has been to study the combination in ~10,000 subjects to rule out rare serious adverse events that could be missed in studies with small sample sizes. Such was done recently in lymphatic filariasis when IDA (ivermectin-diethylcarbamazine-albendazole) was being considered and the process has been described: Weil GJ, Fischer PU, Krentel A. Lessons from Large-Scale Tolerability and Acceptability Studies of Triple Drug Mass Drug Administration Performed to Support Policy Change and Accelerate Elimination of Lymphatic Filariasis. Am J Trop Med Hyg. 2022 Mar 15;106(5_Suppl):13-17. Perhaps the authors could comment on the target policy profile of ivermectin-albendazole-azithromycin and whether a more formal assessment should be undertaken for this combination.

Reviewer #2: I have reviewed the manuscript titled “Pharmacodynamics, feasibility and safety of co-administering azithromycin, albendazole, and ivermectin during mass drug administration: a review”. The manuscript provides a comprehensive review of combination drug therapy utilized in mass drug administration. 

Major comments:

1. Pharmacodynamics in the title. The manuscript focuses more on safety, side effects and pharmacokinetic drug interactions. Pharmacodynamics is not a major focus of the manuscript and I would recommend removal from the title.

2. It is not clear what side affects were evaluated. It would be useful to describe the side effects reported/evaluated if there were consistent across studies. Perhaps listing major side effects and frequency in tabular form would be helpful.

3. There is no dosing information provided. It would be useful to include doses throughout, or perhaps a table with dose information included.

PLOS authors have the option to publish the peer review history of their article (what does this mean?). If published, this will include your full peer review and any attached files.

Reviewer #1: No

Reviewer #2: No

Figure Files:

Data Requirements:

Reproducibility:

References

---

## [Editor Report · Decision Letter 1]

31 May 2023

Dear Dr. Marks,

We are pleased to inform you that your manuscript 'Pharmacokinetics, feasibility and safety of co-administering azithromycin, albendazole, and ivermectin during mass drug administration: a review' has been provisionally accepted for publication in PLOS Neglected Tropical Diseases.

Best regards,

Timothy G. Geary, PhD

Guest Editor

Mathieu Picardeau

Section Editor

The authors have fully resolved the concerns raised by the reviewers and are thanked sincerely for their positive and constructive responses. Ihe manuscript has been improved and meets the standards associated with publication in PLoS-NTDs.

---

## [Editor Report · Acceptance letter]

5 Jun 2023

Dear Dr. Marks,

We are delighted to inform you that your manuscript, "Pharmacokinetics, feasibility and safety of co-administering azithromycin, albendazole, and ivermectin during mass drug administration: a review," has been formally accepted for publication in PLOS Neglected Tropical Diseases.

Best regards,

Shaden Kamhawi

co-Editor-in-Chief

Paul Brindley

co-Editor-in-Chief
